# A Deep-Learning Based Approach to Accelerate Groundtruth Generation for Biomarker Status Identification in Chromogenic Duplex Images

**Satarupa Mukherjee**                                    SATARUPA.MUKHERJEE@ROCHE.COM
**Qinle Ba**                                                      QINLE.BA@ROCHE.COM
**Jim Martin**                              JIM.MARTIN@CONTRACTORS.ROCHE.COM
**Yao Nie**                                                        YAO.NIE@ROCHE.COM
*Roche Sequencing Solutions, Santa Clara, CA, USA*

**Editors:** Accepted for MIDL 2023

## Abstract

Immunohistochemistry based companion diagnosis relies on the examination of single biomarkers for patient stratification. However, recent years have seen an increasing need to characterize the interactions among biomarkers in the tumor microenvironment. To this end, chromogenic multiplexing immunohistochemistry (mIHC) serves as a promising solution, which enables simultaneous detection of multiple biomarkers in the same tissue sections. To automate whole-slide scoring for mIHC, a crucial analysis step involves the identification of cell locations along with their biomarker staining status (presence/absence of positive staining signals), which we call biomarker status identification. However, developing algorithms for such analysis, especially deep-learning (DL) models, often requires manual labeling at the cell-level, which is time-consuming and resource-intensive. Here, we present a DL based method to accelerate groundtruth label generation for chromogenic duplex (tissue samples stained with two biomarkers) images. We first generated approximate cell labels and then developed a DL based interactive segmentation system to efficiently refine the cell labels. Our method avoided extensive manual labeling and reduced the time of label generation to 50%-25% of manual labeling, while achieving <5% error rate in pathologist review.
**Keywords:** Deep Learning, Biomarker Status, Accelerated Groundtruth Generation

## 1. Introduction

Due to the complex color blending effects (Figure 1(a)) in chromogenic mIHC images, it is challenging for pathologists to visually decouple the staining intensities from multiple biomarkers and thus they cannot reliably interpret biomarker status from these images. One promising solution to this challenge is to develop quantitative analysis algorithms based on deep-learning, to identify the biomarker staining of each cell type of interest and then assemble cell-level results into whole-slide scoring for biomarker status.

Here, we present a novel approach to generate cell-level labels for chromogenic duplex assays. Our method ensures the validity of the obtained labels, while avoiding extensive manual labeling by expert pathologists, significantly reducing labeling time. This method combined approximate cell-level labeling and a generalized DL-based interactive tissue segmentation. We validated the proposed method with a duplex assay for PDL1 (Programmed death-ligand) and CK7 (Cytokeratin). Notably, the proposed method can be readily applied to any other duplex assays.

## 2. Methodology

**Approximate cell labels -** We aimed to label the pixel location of the nucleus center and the biomarker presence/absence for each cell, targeting five classes: (i) PDL1+CK7+ tumor cells, (ii) PDL1+CK7- tumor cells, (iii) PDL1-CK7+ tumor cells, (iv) PDL1-CK7- tumor cells and (v) Other cells. 40 field of views (FOVs) of size 600x600 pixels at 20x magnification were selected by a pathologist from 4 Tamra-PDL1/Dabsyl-CK7 duplex slides (lung cancer) to cover a diverse range of biomarker staining intensities. We used HALO (Indica Labs HALO image analysis platform) for initial stain unmixing, followed by tissue segmentation and biomarker status identification. The tissue segmentation was performed for three classes: (i) Tumor, (ii) Stroma and (iii) Other. The biomarker status identification was performed to locate and classify cells into the following four types: (i) PDL1+CK7+ (ii) PDL1+CK7- (iii) PDL1-CK7+ (iv) PDL1-CK7-. Instead of HALO, any other machine learning based interface could also be used for generating these approximate cell labels from stain unmixed singleplex images.

**Interactive tissue segmentation -** We observed inadequate performance of HALO tissue segmentation and thus developed an interactive segmentation system, inspired by (Sofiiuk et al., 2022). We first trained a DL model that learnt to respond to user input and then developed a GUI to enable users to provide input (mouse clicks) to the model at test time. Unlike existing DL-based interactive segmentation models (Sofiiuk et al., 2022, 2020), which segment one target class at a time (binary segmentation), we designed a three-class model by adding an additional class of user input clicks. We trained our model with HighResolutionNet (Wang et al., 2020) as backbone on the Semantic Boundaries Dataset (SBD) (Hariharan et al., 2011) containing 11355 images (8498 for training; 2857 for validation). Three-class model was preferred because (1) pathologists requested that we show the generated cell labels in the aforementioned three types of regions separately to assist their review and (2) binary models required two rounds of independent segmentation for three types of regions, leading to ambiguity in the mask merging phase.

**Refining approximate cell labels with tissue masks -** We first identified the non-tumor cells located in the non-tumor regions ("Stroma" and "Other" in segmentation masks) that were stained positive for the biomarkers and were thus erroneously labeled as tumor cells in approximate cell labels. These cells included CK7+ and PDL1+ non-tumor cells. We re-labeled these identified cells as the fifth cell class, "Other". In general, such a filtering approach can be leveraged to refine approximate cell labels as needed.

## 3. Results

We observed erroneous segmentation of tumor regions with HALO as well as errors in the approximate cell labeling: (1) macrophages with moderate/strong membrane staining were detected as PDL1+ tumor cells; (2) benign epithelial cells with positive Dabsyl staining were detected as CK7+ tumor cells; (3) many cells in stroma regions with positive Tamra staining were detected as PDL1+ tumor cells; (4) some necrotic cells with positive Tamra staining in the necrotic regions were incorrectly detected as PDL1+ tumor cells.

With the designed interactive segmentation system, we generated accurate tissue segmentation masks (Figure 1(b)) with only a few clicks per tissue class. We found that this system, while trained with natural scene images, could be generalized to chromogenic

mIHC, because it was trained to respond to user input guided by simulated user clicks targeting arbitrarily selected classes of regions. With such tissue masks, incorrect cell labels for macrophages, CK7+ cells in non-tumor regions, PDL1+ cells in the stroma and necrotic regions were re-labeled as "Other" cell types (Figure 1(c)).

To ensure the validity of the cell labels, we first performed stain unmixing (Ruifrok and Johnston, 2001) of the duplex images to generate synthetic CK7 and PDL1 singleplex images respectively, followed by pathologist scoring within the tumor regions in these images. Three pathologists provided scores and their consensus scores (median of their scores) were compared with the groundtruth scores from the corresponding cell labels, as shown in Figure 2, where vertical bars indicate range of pathologist scores. We observed that the quantification from the generated labels aligned well with pathologists' scores, demonstrating the effectiveness of the proposed cell-level label generation method. With this labeling approach, it only took around 15-20 minutes to label an FOV of 600x600 pixels in size, whereas manual annotation took around 45 minutes to 1 hour. With the generated labels, we were able to develop a UNet-based (Ronneberger et al., 2001) model for biomarker status identification at cell-level with >90% accuracy, which was confirmed by 3 pathologists.

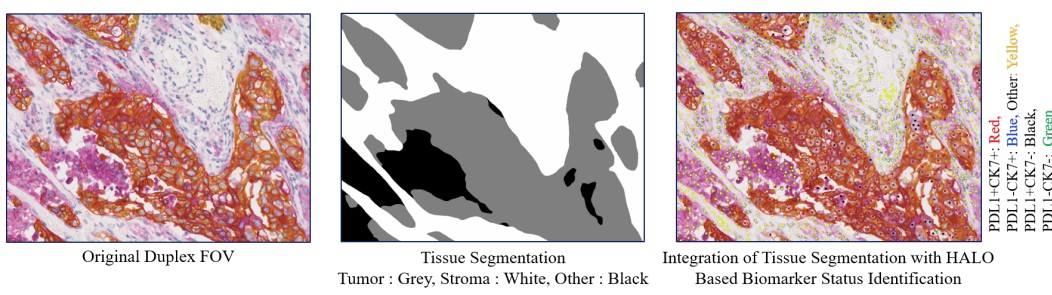

Original Duplex FOV     Tissue Segmentation
Tumor : Grey, Stroma : White, Other : Black     Integration of Tissue Segmentation with HALO Based Biomarker Status Identification

Figure 1: (a) An Example FOV (b) Tissue Segmentation Mask (c) Refined Cell Labels

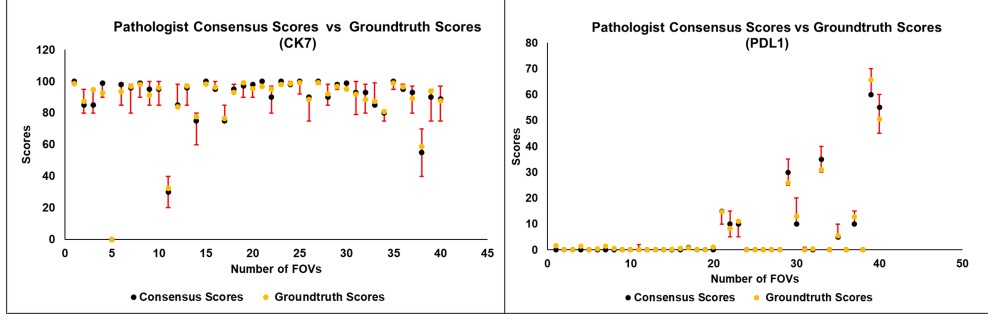

Figure 2: Comparison between Pathologist Scores and Groundtruth Scores

## 4. Conclusion

We have developed a DL-based method for accelerating cell-level labeling with minimal manual input. We first generate approximate cell labels and then develop a DL-based interactive segmentation system to efficiently refine the cell labels. Our labeling approach is highly effective, efficient and readily applicable to various multiplex assays.

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
