# OpenReview forum: "A Deep-Learning Based Approach to Accelerate Groundtruth Generation for Biomarker Status Identification in Chromogenic Duplex Images"
_MIDL.io/2023/Short_Paper_Track — MIDL 2023 Short paper track Poster_

### Official Review · Reviewer_rBet · 2023-04-22

**Rating:** 6
**Confidence:** 5

**Review:**

This paper proposed a deep learning Based method to accelerate groundtruth generation for biomarker status identification in chromogenic duplex images. Experimental results demonstrate significant improvement in reducing the time while achieving competitve performance. This shows potential in real-world setting. The evaluation results under a real clinical setting could further possibly enhance the promise of the method.

---

### Official Review · Reviewer_SuH1 · 2023-04-24
**A Deep-Learning Based Approach to Accelerate Groundtruth Generation for Biomarker Status Identification in Chromogenic Duplex Images**

**Rating:** 6
**Confidence:** 4

**Review:**

This paper presents an approach to annotate cells in histopathology slides of lung cancer stained with multiplexed immunohistochemistry (mIHC).
The goal is to speed up manual annotations while keeping good performance at cell classification.
The HALO platform is used throughout the paper, and equipped with a semi-automatic interactive tissue segmentation method, to fix misclassification of some cell types.
The result should be used to quantify the status of some biomarkers guiding treatment options in oncology, as for example the PDL1 status in tumor cells.

PROS
This is a relevant topic as manual annotations are always tedious and time consuming, especially in histopathology.
The method seems fairly effective based on the presented validation with data from 3 pathologists and their consensus.
Authors claim their method is flexible and can be used also with other combinations of software, not only relying on HALO.

CONS
I find the way the method is presented a bit unclear, and the way I understand it is that the main contribution is the introduction of an interactive tissue segmentation method to fix classification errors made by the HALO (or similar) software.
At the same time, they mention that other software could be used, but it would have been nice to mention a few from the literature.
The part mentioning biomarker quantification only appears in the last sentence of the results, where they mentioned using UNet, but it is unclear how it was trained and for what purpose.